# Regulation of 4-HNE via SMARCA4 Is Associated with Worse Clinical Outcomes in Hepatocellular Carcinoma

**DOI:** 10.3390/biomedicines11082278

**Published:** 2023-08-16

**Authors:** Shiori Watabe, Yukari Aruga, Ryoko Kato, Genji Kawade, Yuki Kubo, Anna Tatsuzawa, Iichiroh Onishi, Yuko Kinowaki, Sachiko Ishibashi, Masumi Ikeda, Yuki Fukawa, Keiichi Akahoshi, Minoru Tanabe, Morito Kurata, Kenichi Ohashi, Masanobu Kitagawa, Kouhei Yamamoto

**Affiliations:** 1Department of Comprehensive Pathology, Graduate School of Medical and Dental Sciences, Tokyo Medical and Dental University, 1-5-45 Yushima, Bunkyo-ku, Tokyo 113-8510, Japan; 2Department of Human Pathology, Graduate School of Medical and Dental Sciences, Tokyo Medical and Dental University, 1-5-45 Yushima, Bunkyo-ku, Tokyo 113-8510, Japan; 3Department of Analytical Information of Clinical Laboratory Medicine, Graduate School of Health Care Science, Bunkyo Gakuin University, 1-19-1 Mukougaoka, Bunkyo-ku, Tokyo 113-8668, Japan; 4Department of Oral Pathology, Graduate School of Medical and Dental Sciences, Tokyo Medical and Dental University, 1-5-45 Yushima, Bunkyo-ku, Tokyo 113-8510, Japan; 5Department of Hepato-Biliary-Pancreatic Surgery, Graduate School of Medical and Dental Sciences, Tokyo Medical and Dental University, 1-5-45 Yushima, Bunkyo-ku, Tokyo 113-8510, Japan

**Keywords:** 4-HNE, SMARCA4, hepatocellular carcinoma, lipid peroxidation, antioxidant enzyme

## Abstract

Accumulation of 4-hydroxynonenal (4-HNE), a marker of lipid peroxidation, has various favorable and unfavorable effects on cancer cells; however, the clinicopathological significance of its accumulation in hepatocellular carcinoma (HCC) and its metabolic pathway remain unknown. This study analyzed 4-HNE accumulation and its clinicopathological significance in HCC. Of the 221 cases, 160 showed relatively low accumulation of 4-HNE in HCC tissues, which was an independent prognostic predictor. No correlation was found between 4-HNE accumulation and the expression of the antioxidant enzymes glutathione peroxidase 4, ferroptosis suppressor protein 1, and guanosine triphosphate cyclohydrolase 1. Therefore, we hypothesized that 4-HNE metabolism is up-regulated in HCC. A database search was focused on the transcriptional regulation of aldo-keto reductases, alcohol dehydrogenases, and glutathione-S-transferases, which are the metabolic enzymes of 4-HNE, and seven candidate transcription factor genes were selected. Among the candidate genes, the knockdown of SWI/SNF-related, matrix-associated, actin-dependent regulator of chromatin, subfamily a, member 4 (SMARCA4) increased 4-HNE accumulation. Immunohistochemical analysis revealed an inverse correlation between 4-HNE accumulation and SMARCA4 expression. These results suggest that SMARCA4 regulates 4-HNE metabolism in HCC. Therefore, targeting SMARCA4 provides a basis for a new therapeutic strategy for HCC via 4-HNE accumulation and increased cytotoxicity.

## 1. Introduction

Hepatocellular carcinoma (HCC) is a major malignancy, accounting for approximately 90% of primary liver cancer cases, and its carcinogenesis has been reported to be associated with hepatitis C virus (HCV) and hepatitis B virus (HBV) infection, alcohol, cirrhosis, and nonalcoholic steatohepatitis. It is the fourth leading cause of cancer-related death worldwide, and, therefore, new cancer therapies are critical to improve the prognosis of patients with HCC [1].

Metabolic changes in cancer cells or metabolic reprogramming occurs during carcinogenesis and cancer progression [2,3], resulting in various abnormalities in metabolites that accumulate in cancer cells [4]. 4-hydroxynonenal (4-HNE) is a lipid peroxide, a marker reflecting the lipid oxidation state, as well as a metabolite with various effects on cells [5]. For instance, 4-HNE reacts with nucleic acids to induce genetic mutations by inhibiting nucleic acid excision repair [6], inducing DNA damage and chromosomal aberrations [7]. It also forms an adduct with p53, a tumor-suppressor gene, and is involved in cancer malignancy [8]. Although 4-HNE inhibits apoptosis [9], its excessive accumulation induces apoptosis [10,11]. It has also been reported that, when nuclear factor erythroid 2–related factor 2 is activated by 4-HNE, it induces the expression of various genes and triggers changes in intracellular signaling [12]. As mentioned above, HNE is cytotoxic, and its accumulation is involved in carcinogenesis [13]. Abnormal accumulation of 4-HNE has been observed in various cancers and other pathological conditions. A higher accumulation of 4-HNE has been reported in nonalcoholic fatty liver disease cases than in controls and is also reported as a prognostic factor in HCV-mediated HCC, where its accumulation is associated with poor prognosis [14,15]. According to previous reports, 4-HNE accumulation is increased in esophageal squamous cell carcinoma, colon adenocarcinoma, lung cancer, and thyroid cancer [16,17,18,19]. Conversely, other studies reported decreased 4-HNE accumulation in colorectal cancer [20] and in clear-cell renal-cell carcinoma and urothelial carcinoma [21]. Changes in the level of 4-HNE accumulation vary depending on the pathological condition and the cancer type.

The metabolic pathway of 4-HNE can be broadly classified into two categories. First, enzymes of antioxidant genes, such as glutathione peroxidase 4 (GPX4), ferroptosis suppressor protein 1 (FSP1), and guanosine triphosphate cyclohydrolase 1 (GCH1) inhibit lipid peroxidation through independent pathways [22,23,24,25,26], resulting in reduced 4-HNE accumulation. Hence, their expression levels influence cancer development, progression, and resistance to chemotherapy [25,27,28]. In HCC, GPX4 overexpression has been reported to be associated with a poor prognosis [29,30]. In contrast, reduced expression of GPX4 and/or FSP1 is associated with poor prognosis in lung squamous cell carcinoma, and the depletion of both enzymes can induce nonapoptotic cell death [31]. Second, the accumulated 4-HNE is metabolized by metabolic enzymes, such as aldo-keto reductase (AKR), alcohol dehydrogenase (ALDH), and glutathione-S-transferase (GST) [10,32], which have multiple isoforms and are regulated by various transcription factors. Exposure to 4-HNE has also been reported to increase the expression of AKR1C in the human neuroblastoma cell line SH-SY5Y [33] and that of glutathione S-transferase alpha 4 (GSTA4), one of the isoforms of GSTs, in colorectal cancer [17]. In various cancers, including HCC, elucidation of the mechanisms of 4-HNE metabolism is important as it contributes to the development of new therapeutic strategies promoting cell death. Although a previous study reported that GSTA4 is overexpressed in HCC, which indicates poor prognosis [34], the correlation of 4-HNE accumulation with clinicopathological factors in cancers, including HCC, and the factors that regulate 4-HNE remain unclear.

In this study, we used immunohistochemical staining to elucidate the clinicopathological significance of 4-HNE accumulation in HCC, and examined the metabolic pathway of 4-HNE using database analysis and in vitro experiments. Our study findings suggest that SMARCA4 is a key regulator of 4-HNE metabolism and has potential as a novel therapeutic target.

## 2. Materials and Methods

### 2.1. Samples

A total of 221 formalin-fixed paraffin-embedded (FFPE) samples of HCC that were resected and diagnosed at the Tokyo Medical and Dental University Hospital between 2008 and 2014 were used in this study. Each FFPE sample comprised of a section containing tumor and non-tumor areas. This study was approved by the ethics committee of Tokyo Medical and Dental University (approval no. M2015-548).

### 2.2. Immunohistochemistry (IHC)

FFPE tissues were sliced into 4-micrometer sections, which were then placed on silane-coated slides. IHC of 4-HNE, GPX4, FSP1, GCH1, and SMARCA4 was performed using the avidin and biotinylated enzyme (ABC) method (VECTASTAIN ABC kit; Vector Laboratories, Newark, CA, USA) or the polymer method (ImmPRESS Reagent kit; Vector Laboratories), and the staining was visualized using Vector 3,3′-Diaminobenzidine Substrate (Vector Laboratories). The details of the staining conditions are presented in Appendix A.

The cytoplasmic expression levels of 4-HNE, GPX4, FSP1, and GST1 were evaluated, compared with those in non-tumor tissues, and scored as follows: 0, weak; 1, comparable; and 2, strong. For SMARCA4, nuclear expression was scored as follows: 1, comparable; 2, partially strong; and 3, strongly positive in whole tumor tissue. All tumor and non-tumor boundaries were identified, and viable tumor cells and non-tumor hepatocyte components of the borderline were compared. The most dominant staining attitude of both areas was used as the result, and the necrotic area was excluded from the evaluation.

### 2.3. Clinicopathological Analysis

Twelve clinicopathological factors were analyzed as follows: age (<70 vs. ≥70 years), sex (male vs. female), alanine transaminase (≤50 vs. >50 IU/L), cirrhosis (yes vs. no), tumor size (≤5 cm vs. >5 cm), number of lesions (single vs. multiple), vascular invasion (yes vs. no), HCV infection (yes vs. no), HBV infection (yes vs. no), differentiation (well vs. moderate vs. poor), stage (stage I, II vs. III, IV), and Ki-67 index (<30% vs. ≥30%).

For univariate and multivariate analyses, 4-HNE accumulation and the expression of GPX4, FSP1, and GCH1 were analyzed.

### 2.4. Database Analysis

For the estimation of the metabolic cascade of 4-HNE, the following online databases were used: Oncomine (https://www.oncomine.org/ (accessed on 15 November 2020)), The Human Protein Atlas (https://www.proteinatlas.org/ (accessed on 31 September 2021) [35]), and Chromatin Immunoprecipitation (ChIP)-Atlas (http://chip-atlas.org/ (accessed on 31 September 2021) [36]). The search criteria for Oncomine were mRNA levels in cancer tissue that were higher than normal (*p* < 0.05), while those for the ChIP-Atlas were fold enrichment ≥ 1, log *p*-value < −1.3, threshold for significance = 50, and distance range from transcription start site (TSS) of −50 ≤ TSS ≤ +50.

### 2.5. Cell Line and Culture

The HCC cell line HepG2 was purchased from the Japanese Collection of Research Bioresources Cell Bank and incubated in Dulbecco’s Modified Eagle’s Medium (Wako Chemicals, Richmond, VA, USA) with 10% fetal bovine serum (Gibco, Waltham, MA, USA) and 1% penicillin and streptomycin (Gibco). The cells were passaged at a ratio of 1:5 every 2–3 days.

### 2.6. Small Interfering RNA (siRNA) Transfection

siRNAs targeting *FOXP1*, *MCRS1*, *PROX1*, *RUVBL1*, *SMARCA4*, *SP1*, and *ZMYM3* were obtained from Thermo Fisher Scientific (Waltham, MA, USA; Appendix A). Non-target sequence siRNA (Thermo Fisher Scientific, 4390843) was used as a control. Using 1 × 10^5^ cells per sample, the cells were transfected via Lipofectamine 3000 (Invitrogen, Waltham, MA, USA) according to the manufacturer’s instructions, and Opti-modified Eagle’s Medium (Gibco) was used. After 72 h, the cells were collected for enzyme-linked immunosorbent assay (ELISA) analysis.

### 2.7. Quantitative Reverse Transcription Polymerase Chain Reaction

RNA was extracted from siRNA-transfected cells using the RNeasy Mini Kit (Qiagen, Hilden, Germany) and was reverse transcribed to complementary DNA (cDNA) using TaqMan reverse transcription reagents (Applied Biosystems, Woburn, MA, USA). Messenger RNA (mRNA) expression was quantified on an ABI PRISM 7900HT Sequence Detection System (Applied Biosystems) using the THUNDERBIRD SYBR qPCR Mix (Toyobo, Tokyo, Japan). In this study, *β-actin*, *FOXP1*, *MCRS1*, *PROX1*, *RUVBL1*, *SMARCA4*, *SP1*, and *ZMYM3* mRNA levels were measured, with *β-actin* as a normalization control. The primer sequences are shown in Appendix A.

### 2.8. ELISA

For sample preparation, 2 × 10⁵ siRNA-transfected cells were suspended in 300 µL radioimmunoprecipitation assay buffer (50 mmol/L Tris-HCl [pH 8.0], 150 mmol/L NaCl, 0.1% *w*/*v* sodium dodecyl sulfate, 1.0% *w*/*v* NP40, and 0.5% *w*/*v* deoxycholic acid) and allowed to rest on ice for 30 min, after which they were sonicated and centrifuged at 9000× *g* for 15 min. The supernatant was collected, and 4-HNE concentration was quantified using the Lipid Peroxidation (4-HNE) Assay Kit (Abcam, Cambridge, UK) according to the manufacturer’s instructions. This experiment was repeated two times.

### 2.9. Western Blotting

The efficiency of SMARCA4 knockdown by siRNA in HepG2 cells was confirmed using Western blotting. Methods previously described [37] were applied to HepG2 in this study. The primary antibodies used in this study were SMARCA4/BRG1 (21634-1-AP) (Proteintech, Chicago, IL, USA) and b-Actin (13E5) (Cell Signaling Technologies, Danvers, MA, USA) at the dilution of 1:1000. Anti-Rabbit IgG, HRP-Linked Whole Ab Donkey (Cytiva, Tokyo, Japan), was used as a secondary antibody at the dilution rate 1:5000.

### 2.10. Statistical Analysis

Correlations between two groups were determined using Fisher’s exact test. Overall survival duration was calculated from the date of diagnosis to the date of death or last follow-up. Kaplan–Meier survival curves were used to estimate overall survival rates, and log-rank tests were used to assess differences in survival between groups. Univariate and multivariate analyses were performed using the log-rank test and the Cox proportional-hazard regression model, respectively. One-way analysis of variance with Dunnett’s test was used for ELISA analysis. All statistical analyses were performed using EZR software (version 4.0.3; Saitama Medical Center, Jichi Medical University, Saitama, Japan). The error bars represent the standard deviation in this study.

## 3. Results

### 3.1. Immunohistochemical Analysis of 4-HNE and Antioxidant Enzymes

Immunostaining for 4-HNE and the antioxidant enzymes GPX4, FSP1, and GCH1 was performed. Representative images of the immunostaining are shown in Figure 1. For 4-HNE, the numbers of cases with scores of 0 (staining intensity of tumor area was weaker than that of the non-tumoral area), 1 (staining intensity of tumor and non-tumoral area were comparable), and 2 (staining intensity of tumor area was stronger than that of the non-tumoral area) were 160 (72.4%), 34 (15.4%), and 27 (12.2%) out of 221 cases, respectively. For GPX4, the numbers of cases with scores of 0, 1, and 2 were 34 (15.4%), 110 (50.0%), and 77 (34.8%), respectively. For FSP1, the numbers of cases with scores of 0, 1, and 2 were 94 (42.5%), 92 (41.6%), and 35 (15.8%), respectively. For GCH1, the numbers of cases with scores of 0, 1, and 2 were 81 (36.7%), 65 (29.4%), and 75 (33.9%), respectively (Appendix A).

### 3.2. Survival and Clinicopathological Analyses with 4-HNE Accumulation and Antioxidant Enzymes

Patients were stratified according to 4-HNE accumulation and the expression levels of GPX4, FSP1, and GCH1, and the correlation between the accumulation or expression levels and overall survival was analyzed (Figure 2a–h). Results showed that 4-HNE accumulation and GPX4 protein expression were correlated with overall survival. In the comparison among the three groups (scores 0, 1, and 2), the prognosis tended to be worse in cases of cancer tissues with less 4-HNE accumulation than in noncancerous tissues (Figure 2a). In contrast, a worse prognosis was observed in cases with higher or less GPX4 expression in cancer tissues than in noncancerous tissues (Figure 2c). The prognosis based on 4-HNE was significantly worse in the score 0 group (*p* = 0.034; Figure 2b), while that based on GPX4 was worse in the score 2 group (*p* = 0.005, Figure 2d). There were no significant differences in the prognosis classified according to the expression of FSP1 or GCH1 (Figure 2e–h).

Next, we analyzed the correlations of 4-HNE accumulation or the expression levels of GPX4, FSP1, and GCH1 with clinicopathological factors in HCC and found that 4-HNE accumulation significantly correlated with the number of lesions (*p* = 0.049), vascular invasion (*p* = 0.002), differentiation (*p* = 0.002), stage (*p* = 0.004), and Ki-67 score (*p* = 0.001), whereas GPX4 expression levels significantly correlated with the presence of cirrhosis (*p* = 0.004) and HCV infection (*p* = 0.016). Meanwhile, FSP1 and GCH1 expression did not show a marked correlation with clinicopathological factors (Appendix A). In addition, univariate analysis of clinicopathological factors, 4-HNE accumulation, and GPX4, FSP1, and GCH1 expression, was performed, and 4-HNE accumulation (*p* = 0.003), GPX4 expression (*p* = 0.005), tumor size (*p* = 0.001), number of lesions (*p* = 0.006), and vascular invasion (*p* = 0.004) were significantly associated with prognosis (Table 1). Multivariate analysis of these factors revealed that tumor size (hazard ratio (HR): 2.386, 95% confidence interval (CI): 1.328–4.286, *p* = 0.004), 4-HNE score 0 (HR: 2.726, 95% CI: 1.127–6.596, *p* = 0.026), and relatively high GPX4 expression (HR: 2.418, 95% CI: 1.379–4.242, *p* = 0.002) were independent predictive factors for poor prognosis. Decreased 4-HNE accumulation may be advantageous for tumor cells to survive; therefore, we conducted a detailed analysis of the factors that contribute to the decrease in 4-HNE.

### 3.3. Correlation between 4-HNE Accumulation and Antioxidant Enzyme Expression

The estimated mechanism for the reduction in 4-HNE in HCC is that antioxidant enzymes are overexpressed and cancel lipid peroxidation, thereby avoiding the consequent generation of 4-HNE and its associated cytotoxicity. Therefore, we examined the relationship between 4-HNE accumulation and the expression of antioxidant enzymes; however, no significant correlation was found between them (Table 2). Next, the relationship between the expression of GPX4, FSP1, and GCH1 with score 2 in each case and the amount of 4-HNE accumulated was analyzed. The percentage of cases with score 2 for any one of the antioxidant enzymes was 60.0% for 4-HNE with score 0 and 60.7% for 4-HNE with score 1 and score 2, with no significant difference (Table 3). These findings suggest that 4-HNE accumulation is reduced in HCC through mechanisms other than the elimination of lipid peroxidation by GPX, FSP1, and GCH1.

### 3.4. Database Search for Upstream Factors of 4-HNE Metabolism

A possible mechanism of 4-HNE metabolism may be involved in the relative reduction in 4-HNE accumulation in HCC. Therefore, the factors that regulate the expression of 4-HNE metabolic enzymes were formulated under several conditions, as shown in the flow chart in Figure 3a. There are 15, 19, and 21 AKR, ALDH, and GST isoforms, respectively, that contribute to 4-HNE metabolism [38,39,40]. Using the chromatin precipitation database ChiP-Atlas, we formulated transcription-related factors that bind specifically to the genes of these metabolic enzymes. Forty candidate genes were extracted as follows: two candidates for AKRs (*ARID3A* and *FOXA2*), nine genes for ALDHs (*FOXA1*, *FOXA2*, *HNF4A*, *HNF4G*, *MAFF*, *MAFK*, *SNRNP70*, *USF2*, and *ZMYM3*), and eighteen genes for GSTs (*CCAR2*, *FOXA1*, *FOXP1*, *GFP*, *HNF4A*, *MAFF*, *MAFK*, *MBD4*, *MCRS1*, *MED1*, *PROX1*, *RXRA*, *RUVBL1*, *SMARCA4*, *SP1*, *TEAD4*, *TWIST1*, and *YY1*). In addition, using Oncomine, a gene expression comparison database, 12 of the above 40 genes were selected based on the condition that they are related to transcription and have higher mRNA expression in HCC than in normal hepatocytes (*FOXA1*, *FOXP1*, *MCRS1*, *MED1*, *PROX1*, *RUVBL1*, *SMARCA4*, *SP1*, *SPI1*, *TEAD4*, *YY1*, and *ZMYM3*). Using The Human Protein Atlas, a comparative database of protein expression, 12 genes that had protein expression in HCC tissues were selected. Finally, seven candidate genes, namely, *FOXP1*, *MCRS1*, *PROX1*, *RUVBL1*, *SMARCA4*, *SP1*, and *ZMYM3*, were identified as transcription factor-related genes associated with 4-HNE metabolism.

### 3.5. In Vitro Validation of Genes Selected via Database Analysis

Seven candidate genes identified by database analysis were examined for their association with 4-HNE accumulation in vitro. Each gene was knocked down using siRNA (Appendix A), and the intracellular accumulation of 4-HNE was evaluated using ELISA. The results showed a significant increase in 4-HNE accumulation following *SMARCA4* knockdown (2.12-fold, *p* = 0.049, Figure 3b). Western blot analysis confirmed SMARCA4 protein expression was down regulated in cells transfected with si SMARCA4 (Figure 3c).

### 3.6. IHC of SMARCA4 and the Relation between Its Expression and 4-HNE Accumulation

Considering ELISA results, SMARCA4 is a candidate transcription factor that regulates the expression of a group of metabolic enzymes; therefore, immunohistochemical analysis of the association between SMARCA4 expression and 4-HNE accumulation in HCC tissues was performed (Figure 4a–c and Table 4). The result showed a significant correlation between the increased expression of SMARCA4 and decreased accumulation of 4-HNE (*p* = 0.041). In addition, patients were stratified according to SMARCA4 expression levels, and the result of their association with the outcome is shown in Figure 4d. When classified into three groups for each score, a trend toward worse prognosis was observed in the group with score 3. The results were divided into two groups: (a) group with score 1 and score 2 and (b) group with score 3. The group with score 3, which showed higher expression of SMARCA4 in whole tumor tissue than in normal hepatocytes, had a significantly poorer prognosis than the group with score 1 and score 2 (*p* < 0.001).

## 4. Discussion

4-HNE is the final metabolite of lipid peroxidation, resulting from the peroxidation of ω6 polyunsaturated fatty acids such as linolenic and arachidonic acids. It is associated with carcinogenesis and cancer progression, has highly reactive C=C and C=O, and forms adducts with intracellular substances such as proteins, nucleic acids, and phospholipids, leading to impaired intracellular signal transduction, genetic mutations, and cytotoxicity [5,41]. This study showed that the accumulation of 4-HNE is relatively low in cancer tissues compared with that in noncancerous tissues, which is associated with a poor prognosis. It has been reported that exposure to low concentrations of 4-HNE promotes cell proliferation, whereas exposure to high concentrations of 4-HNE inhibits proliferation. Moreover, higher concentrations of 4-HNE induce cell death, such as apoptosis and necrosis [6,42]. In cases with lower 4-HNE accumulation in HCC, 4-HNE may be less cytotoxic and favor proliferation and survival. On the other hand, a high accumulation of 4-HNE also has a poor prognosis, but in these cases, it may have acquired some form of viability that exceeds the cytotoxicity of 4-HNE itself, which may include the influence of other factors not examined in this study.

An inverse correlation between GPX4 expression and accumulation of 8-hydroxy-2′-deoxyguanosine, the end-product of oxidation, has been observed in diffuse large B-cell lymphoma (DLBCL) [43], whereas GPX4 expression is upregulated, and oxidative stress is reduced in colon cancer [44]. However, no significant inverse correlation was observed between 4-HNE accumulation and antioxidant enzyme expression in HCC, which indicates that 4-HNE accumulation in HCC may be reduced by means other than the high expression of antioxidant enzymes and elimination of lipid peroxidation in cases with relatively low 4-HNE accumulation. Among the candidate factors regulating the 4-HNE-metabolizing enzymes AKRs, ALDHs, and GSTs, SWI/SNF-related, matrix-associated, actin-dependent regulator of chromatin, subfamily a, member 4 (SMARCA4) is suggested to regulate the metabolism of 4-HNE in HCC. SMARCA4 is involved in chromatin remodeling and promotes or inhibits the expression of several genes by forming the BRM-associated factor complex [45]. Since SMARCA4-deficient tumors in thoracic malignancies were reported in 2015 [46], SMARCA4 expression and its genetic mutations have received attention in various cancers [47,48,49,50]. SMARCA4 acts as a tumor suppressor in hypercalcemic small cell carcinoma of the ovary, and its deficiency has been reported to be involved in carcinogenesis [47]. SMARCA4 not only acts as a tumor suppressor but also cooperates with p53 loss and Kras activation in mice, and SMARCA4 enhances oxidative phosphorylation [48]. In contrast, upregulation of SMARCA4 is observed in HCC, in which it promotes cell proliferation and progression, and a worse clinical outcome was observed in the group with higher SMARCA4 expression [49,50]. Furthermore, SMARCA4 expression, in concert with c-MET and NRAS^v12^ mutations, is involved in HCC carcinogenesis [49], whereas its inhibition reduces cell migration [51]. A database search (http://chip-atlas.org/, accessed on 31 September 2021) showed that SMARCA4 regulates GSTA4, a 4-HNE metabolic enzyme. Of the GSTs, GSTA4 is known to have a particularly high metabolic capacity for 4-HNE [52]. GSTs contribute to the metabolism of 4-HNE either by forming a complex with reduced glutathione and being excreted through efflux transporters (MRP and RLIP76) or via further metabolism of this complex by AKR and ALDH [10].

In this study, SMARCA4 negatively regulated 4-HNE accumulation in HCC, suggesting that it positively regulates the expression of 4-HNE metabolic enzymes and contributes to 4-HNE metabolism and its decrease in cancer tissues. This reduces the cytotoxicity of 4-HNE and may favor the survival of HCC cells. Furthermore, SMARCA4 upregulated the expression of HO-1, an antioxidant enzyme [53], which may reduce intracellular oxidative stress. Therefore, targeting SMARCA4 and its downstream metabolic enzymes is expected to lead to the development of novel therapeutic strategies due to 4-HNE accumulation and its associated cytotoxicity and oxidative stress accumulation.

In addition, a poor prognosis was observed in the group with high expression of the antioxidant enzyme GPX4, suggesting that lipid peroxidation was suppressed in this group. In turn, this suggests that low 4-HNE expression may be partly due to the overexpression of GPX4, which is a key negative regulator of ferroptosis (nonapoptotic cell death related to lipid peroxidation) that has recently attracted attention [23]. The overexpression of GPX4 may suppress ferroptosis, thus favoring cancer cell survival. It is advantageous to establish the molecular basis for a novel antitumor strategy by inducing ferroptosis in HCC with high GPX4 expression.

Despite these findings, this study has several limitations. The expression levels of GPX4, FSP1, and GCH1 in HCC differed from case to case; however, we were unable to investigate the molecular mechanisms that regulate this in the present study. Overexpression of GPX4 in DLBCL has been shown to upregulate *SECISBP2*, which regulates GPX4 expression, and its expression may be controlled by a similar mechanism in HCC [54]. Moreover, we found that 4-HNE reduction serves as a poor prognostic factor in HCC tissues and is detected in SMARCA4 knockdown cells in vitro; however, the current study lacks in vivo support. Therefore, additional in vivo experiments are necessary to support our claim. In addition, the mechanism of SMARCA4 overexpression, which is a poor prognostic predictor of HCC, remains unclear. Furthermore, the metabolic enzymes related to 4-HNE that are regulated by SMARCA4 have not been fully investigated, and further studies are needed to determine whether SMARCA4 regulates GSTA4 and other metabolic enzymes. In addition to SMARCA4, the possibility that FOXP1 also regulates 4-HNE is undeniable, although ELISA experiments showed no statistical differences. The standard deviation was large due to the small number of repeated experiments in this study. Therefore, further studies are needed to obtain more reliable results.

## 5. Conclusions

Collectively, our findings elucidate the clinicopathological significance of 4-HNE in HCC and the molecular mechanisms underlying the 4-HNE metabolic pathway. Future studies on the details of SMARCA4-mediated metabolism of 4-HNE and examination of preclinical models may help establish evidence for SMARCA4 as a therapeutic target and improve the prognosis of patients with HCC.

## Figures and Tables

**Figure 1 biomedicines-11-02278-f001:**
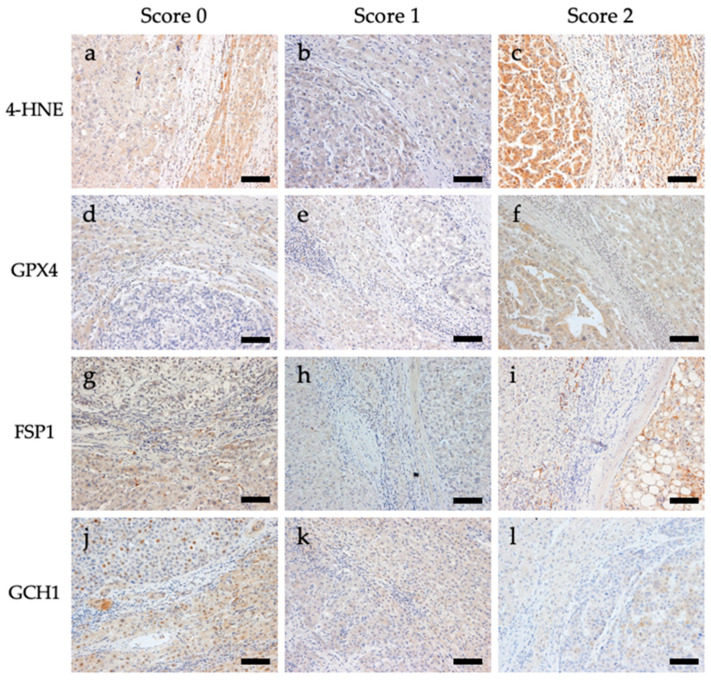
Representative images of immunohistochemical staining of 4-hydroxynonenal (4-HNE), glutathione peroxidase 4 (GPX4), ferroptosis suppressor protein 1 (FSP1), and guanosine triphosphate cyclohydrolase 1 (GCH1) in tissues from patients with hepatocellular carcinoma. All images are shown at a magnification of 200×, and the scale bars indicate 50 µm. (**a**) 4-HNE score 0, (**b**) 4-HNE score 1, (**c**) 4-HNE score 2, (**d**) GPX4 score 0, (**e**) GPX4 score 1, (**f**) GPX4 score 2, (**g**) FSP1 score 0, (**h**) FSP1 score 1, (**i**) FSP1 score 2, (**j**) GCH1 score 0, (**k**) GCH1 score 1, and (**l**) GCH1 score 2.

**Figure 2 biomedicines-11-02278-f002:**
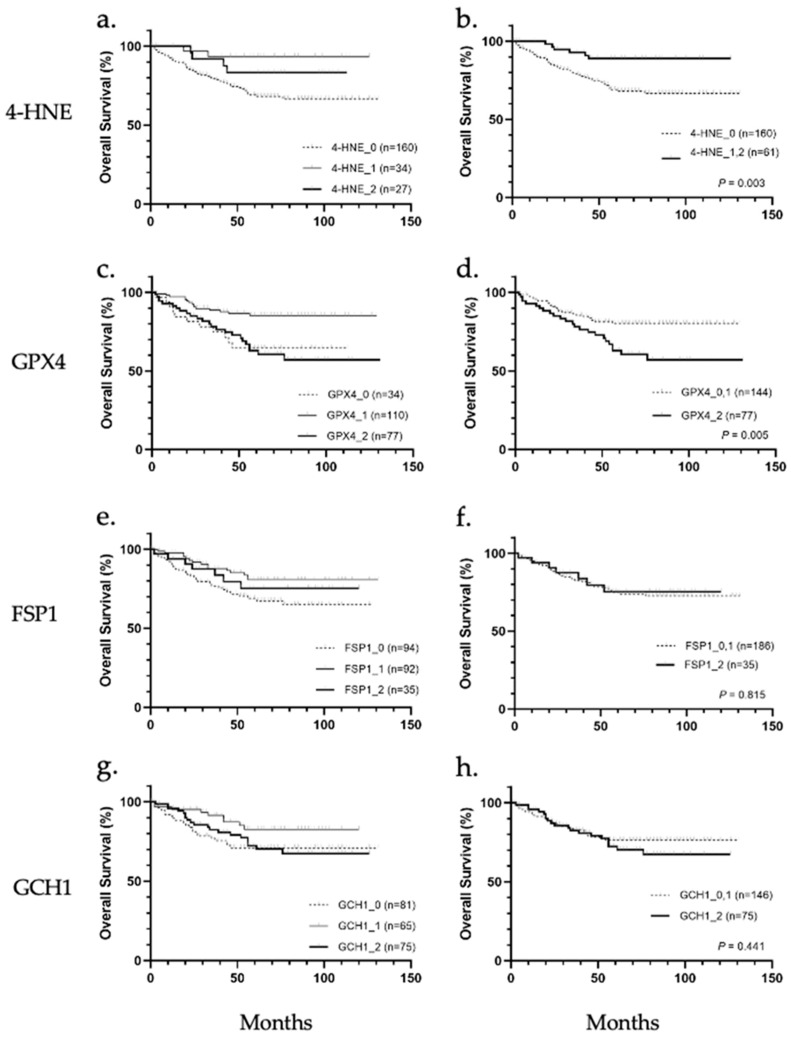
Overall survival analysis of hepatocellular carcinoma stratified according to immunohistochemical staining of 4-hydroxynonenal (4-HNE), glutathione peroxidase 4 (GPX4), ferroptosis suppressor protein 1 (FSP1), and guanosine triphosphate cyclohydrolase 1 (GCH1). (**a**) Classification into three groups based on 4-HNE staining intensity showed no significant association with overall survival. (**b**) 4-HNE score 0 group showed significantly worse prognosis than 4-HNE score 1 and score 2 groups (*p* = 0.003). (**c**) Classification into three groups based on GPX4 staining intensity showed no significant association with overall survival. (**d**) GPX4 score 2 group showed significantly worse prognosis than the GPX4 score 0 and score 1 group (*p* = 0.005). (**e**) Classification into three groups based on FSP1 staining intensity. (**f**) None of the groups (FSP1 score 0, 1, and 2) were significantly associated with overall survival. (**g**) Classification into three groups based on GCH1 staining intensity. (**h**) None of the groups (GCH1 score 0, 1, and 2) were significantly associated with overall survival.

**Figure 3 biomedicines-11-02278-f003:**
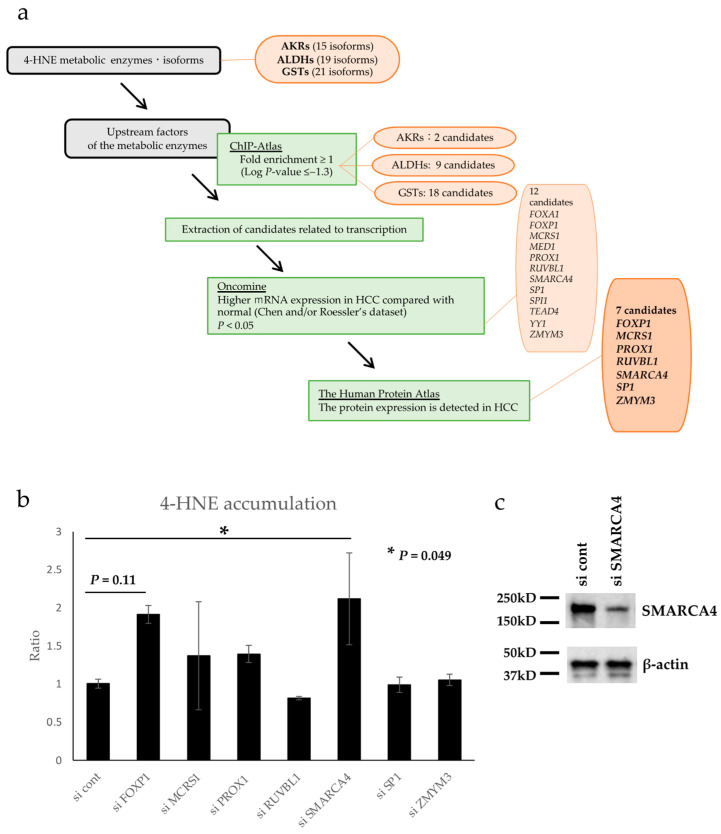
Flowchart for database analysis and 4-hydroxynonenal (4-HNE) quantification by enzyme-linked immunosorbent assay (ELISA). (**a**) The database was analyzed as illustrated in the flow chart. Seven candidate genes were finally identified. (**b**) Intracellular 4-HNE accumulation was quantified by ELISA, and the knockdown of *SMARCA4* significantly increased 4-HNE accumulation (*p* < 0.05). The error bars represent standard deviation. Abbreviations: AKR, aldo-keto reductase; ALDH, alcohol dehydrogenase; GST, glutathione-S-transferase; HCC, Hepatocellular carcinoma. (**c**) Western blot analysis of SMARCA4 confirmed down-regulation of its expression with si SMARCA4.

**Figure 4 biomedicines-11-02278-f004:**
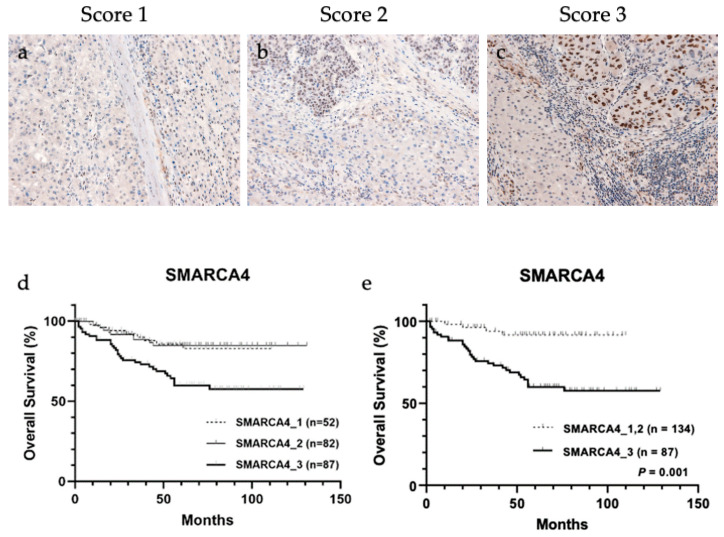
Representative images of immunohistochemical staining of SWI/SNF-related, matrix-associated, actin-dependent regulator of chromatin, subfamily a, member 4 (SMARCA4) in tissues from patients with hepatocellular carcinoma (HCC). (**a**) Score 1, (**b**) score 2, and (**c**) score 3. Overall survival analysis of HCC based on SMARCA4 staining intensity. (**d**) Classification into three groups based on SMARCA4 staining intensity showed no significant association with overall survival. (**e**) SMARCA4 score 3 group showed significantly worse prognosis than SMARCA4 score 1 and score 2 groups (*p* < 0.001).

**Table 1 biomedicines-11-02278-t001:** Univariate and multivariate analyses.

Variable		Number of Patients	*p*-Value By Log-Rank Test	HR ^a^	95% CI ^b^	*p*-Value by Cox Proportional Hazard
Age	<70 y ≥70 y	95 126	0.673			
Sex	Male Female	164 57	0.198			
ALT	≤50 IU/L >50 IU/L	152 69	0.536			
Cirrhosis	− +	130 91	0.763			
Tumor size	≤5 cm >5 cm	146 75	**0.001**	2.386	1.328–4.286	**0.004**
Lesion	Single Multiple	155 66	**0.006**	1.991	0.981–4.040	0.057
Vascular invasion	− +	127 94	**0.004**	1.490	0.619–3.586	0.374
HCV infection	− +	120 101	0.827			
HBV infection	− +	183 38	0.521			
Differentiation	Well Moderate Poor	60 118 43	**0.005**	1.525	0.916–2.541	0.105
Stage	I, II III, IV	99 122	**0.006**	0.816	0.284–2.344	0.706
Ki-67	<30% ≥30%	137 84	**0.019**	0.863	0.433–1.719	0.676
4-HNE	Score 1, 2 Score 0	61 160	**0.003**	2.726	1.127–6.596	**0.026**
GPX4	Score 0, 1 Score 2	144 77	**0.005**	2.418	1.379–4.242	**0.002**
FSP1	Score 0, 1 Score 2	186 35	0.815			
GCH1	Score 0, 1 Score 2	146 75	0.440			

Abbreviations: ^a^ HR: Hazard ratio, ^b^ CI: Confidence interval.

**Table 2 biomedicines-11-02278-t002:** Correlation between 4-hydroxynonenal accumulation and the expression of antioxidants against lipid peroxidation.

GPX4					
		**Score 0, 1**	**Score 2**	**% (GPX4 Score 2)**	***p*-value**
4-HNE	Score 0	108	52	32.5	0.270
	Score 1, 2	36	25	41.0	
**FSP1**					
		**Score 0, 1**	**Score 2**	**% (FSP1 Score 2)**	** *p* ** **-value**
4-HNE	Score 0	138	22	13.8	0.215
	Score 1, 2	48	13	21.3	
**GCH1**					
		**Score 0, 1**	**Score 2**	**% (GCH1 Score 2)**	** *p* ** **-value**
4-HNE	Score 0	104	56	35.0	0.636
	Score 1, 2	42	19	31.1	

**Table 3 biomedicines-11-02278-t003:** Correlation between 4-hydroxynonenal accumulation and the number of the high expression of antioxidants for lipid peroxidation.

		The Number of Score 2 in GPX4, FSP1, or GCH1 Per Sample					
		0 ^a^	1 ^b^	2 ^c^	3 ^d^	% ((b + c + d)/(a + b + c + d)) × 100	*p*-Value
4-HNE	Score 0	64	66	26	4	60.0	0.406
	Score 1, 2	24	20	14	3	60.7	

**Table 4 biomedicines-11-02278-t004:** Correlation between 4-hydroxynonenal accumulation and the expression of SWI/SNF-related, matrix-associated, actin-dependent regulator of chromatin, subfamily a, member 4.

		SMARCA4			
		Score 1	Score 2	Score 3	*p*-Value
4-HNE	Score 0	34	55	71	0.041
	Score 1, 2	18	27	16	

## Data Availability

The data presented in this study are available from the corresponding author upon reasonable request.

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
