# Peer review of "Regulation of 4-HNE via SMARCA4 Is Associated with Worse Clinical Outcomes in Hepatocellular Carcinoma"

_biomedicines, 2023, doi:10.3390/biomedicines11082278_

Round 1

Reviewer 1 Report (Previous Reviewer 2)

This manuscript "Regulation of 4-HNE via SMARCA4 is Associated with Worse Clinical Outcomes in Hepatocellular Carcinoma" is overall well presented the results are comprehensive and the topic is also of scientific interest. Only one clarification the authors in material and methods have inserted "Cell Line and Culture" naming the HePG2, but in all the work there is not an experiment that includes the HePG2 and the reference 37 which should clarify is actually referred to ovarian carcinoma .It would be appropriate to clarify this point

Author Response

Reviewer 1

This manuscript "Regulation of 4-HNE via SMARCA4 is Associated with Worse Clinical Outcomes in Hepatocellular Carcinoma" is overall well presented the results are comprehensive and the topic is also of scientific interest.

- Only one clarification the authors in material and methods have inserted "Cell Line and Culture" naming the HePG2, but in all the work there is not an experiment that includes the HePG2 and the reference 37 which should clarify is actually referred to ovarian carcinoma. It would be appropriate to clarify this point

(Response)

I really appreciate your valuable feedback. As you pointed out, the cells used in this study are HepG2, not ovarian carcinoma cells. We performed the experiment by adapting the method of western blot described in paper 37 to HepG2. I have described this in the Materials & Methods section of the text.

Reviewer 2 Report (Previous Reviewer 1)

The submission by Shiori Watabe et al entitled "Regulation of 4-HNE via SMARCA4 is Associated with Worse 2 Clinical Outcomes in Hepatocellular Carcinoma" uses IHC of 4-HNE, GPX4, FSP1 and GCH1 to perform a survival analysis for hepatocellular carcinoma and, having found an association between 4-HNE expression and survival, then infer a role for SMARCA4.

In Fig. 2a, the expression of 4HNE is both highest and lowest in the two groups with reduced survival.  Patients with 4HNE expression equivalent to normal tissue have the best survival.  Can the authors explain why there is no dose-dependent effect?

The weakest point of the submission is the siRNA knockdown in Figure 3b.  Assuming the n per group is the same, it is hard to see how siSMARCA4 has a p value of 0.049 and siFOXP1 has a p value of 0.11.  It is not immediately evident which statistical test was used.  The authors need to strengthen this element of the paper.

Expression of English is fine.

Author Response

(note1 )
In Fig. 2a, the expression of 4HNE is both highest and lowest in the two groups with reduced survival.  Patients with 4HNE expression equivalent to normal tissue have the best survival.  Can the authors explain why there is no dose-dependent effect?

(Response)

 I really appreciate your valuable insights. Different mechanisms may be at work to cause the no-dose dependent effect.

 4-HNE is considered to have a positive effect on cell proliferation and 4-HNE itself is not cytotoxic at low concentrations, so this means cancer cells have advantage for cell proliferation, resulting in poor overall survival. 

 On the other hand, a high accumulation of 4-HNE also has a poor prognosis, but in these cases, it may have acquired some form of viability that exceeds the cytotoxicity of 4-HNE itself, which may include the influence of other factors not examined in this study. Therefore, the same level of non-cancer tissue is considered to be a low malignancy case with neither effect.The above is also described in the Discussion part of the text.

(note 2)

 The weakest point of the submission is the siRNA knockdown in Figure 3b.  Assuming the n per group is the same, it is hard to see how siSMARCA4 has a p value of 0.049 and siFOXP1 has a p value of 0.11.  It is not immediately evident which statistical test was used.  The authors need to strengthen this element of the paper.

(Response)

The experiment was repeated two times in each sample, which was described in Materials & Methods in the text.

 The statistical analysis method was not clear in the previous text, so we added a note in the Materials & Methods section stating that the Dunnett’s method was used. The results of the statistical analysis are included in the attached Word file, so please check it.

Round 2

Reviewer 2 Report (Previous Reviewer 1)

The authors have addressed the points raised in the initial review.

In Figure 3b, the p value givent for control vs siFOXP1 is 0.11 however, the extra statistical detail provided indicates that the p value is 0.1165 which should be rounded to 0.12.

Overall, Figure 3b still does not convince, given the wide error bars for siSMARCA4, however statistical tests can sometimes generate unusual values.

Author Response

In Figure 3b, the p value givent for control vs siFOXP1 is 0.11 however, the extra statistical detail provided indicates that the p value is 0.1165 which should be rounded to 0.12.

(response)

I am very sorry for the mistake. You were definitely right.

Thank you for pointing this out, I have corrected the values listed in Manuscript and Figure.

This manuscript is a resubmission of an earlier submission. The following is a list of the peer review reports and author responses from that submission.

Round 1

Reviewer 1 Report

I have read with interest the submission entitled "Regulation of 4-HNE ... Hepatocellular Carcinoma" by Watabe S. et al.  This manuscript describes an association of 4HNE in liver cancer cells with patient survival and a potential mechanism of 4HNE metabolism by SMARCA4A.  
Increased 4HNE was observed to be associated with better survival.  
SMARCA4A was postulated to decrease 4HNE (based on in vitro siRNA knockdown).
Lower SMARCA4A was observed in tumour tissue from patients with longer survival, presumably leading to increased 4HNE (as implied by Table 4).  Other aspects (antioxidants) were also considered.

Generally speaking the writing is of sufficient quality and the figures/tables clearly represent the data.

Minor points
1) Ln 171-179.  These data would best be represented by a table or graph rather than text description.

2) Fig. 2a.  The expression of 4HNE is both highest and lowest in the two groups with reduced survival.  Patients with 4HNE expression equivalent to normal tissue have the best survival.  Can the authors explain why there is no dose-dependent effect?  A similar effect is observed with GPX4.

3)Ln 156 state g force instead of rpm.

4)Ln 26. "Database search..."  this sentence has no context and appears unrelated to the previous sentence.

5) Ln 85.  "In this study..." New paragraph for conclusion of the Intro.

Major point
6) Fig. 3b.  The error bars are not defined - are they standard error or standard deviation?  The control used for the siRNA experiments is not defined.  Ideally this should be the scrambled sequence of the gene being knocked down.

Given the high variation of SMARCA4A, it is difficult to see how si-FOXP1 is not also statistically significantly different to the control, all things being equal, as the variation is far less and the average is similar to SMARCA4A.  The authors need to improve up on this finding with complimentary experiments or repeat the experiment to increase n.  As it stands, this Figure is not convincing.

English quality is generally fine.  There are a few typos.

Reviewer 2 Report

The authors of the manuscript "Regulation of 4-HNE via SMARCA4 is Associated with Worse Clinical Outcomes in Hepatocellular Carcinoma" address an interesting issue, correlating a lipid metabolite typical of HCC with a protein involved in its metabolism. The data in the manuscript are mainly obtained with immunohistochemical studies on FFPE Tissues and with the help of databases. The results are interesting as they have identified a new potential therapeutic target for HCC.

-It is not clear how they identify 4-HNE in FFPE Tissues to accurately report the methodology.

- In addition to the RT-PCR (supplementary dates), I would also recommend an immunoblotting with ab SMARCA4 of the HePG2 cellular samples to be included in the manuscript and not in the supplementary. This type of evidence is essential to give more credibility to the results presented by the authors.